# Major Pathogens Affecting Carob in the Mediterranean Basin: Current Knowledge and Outlook in Italy

**DOI:** 10.3390/pathogens12111357

**Published:** 2023-11-15

**Authors:** Ivana Castello, Giancarlo Polizzi, Alessandro Vitale

**Affiliations:** Dipartimento di Agricoltura, Alimentazione e Ambiente (Di3A), University of Catania, Via S. Sofia 100, 95123 Catania, Italy; castelloivana75@gmail.com (I.C.); gpolizzi@unict.it (G.P.)

**Keywords:** *Ceratonia siliqua*, powdery mildew, cercospora leaf spot, wood decay fungi, *Xylosandrus compactus*, *Fusarium solani*

## Abstract

The main pathogens affecting the carob (*Ceratonia siliqua*) tree in the Mediterranean basin are described in this overview. The most widespread diseases periodically occurring in carob orchards are powdery mildew (*Pseudoidium ceratoniae*) and cercospora leaf spot (*Pseudocercospora ceratoniae*). The causal agents of “black leaf spots” (e.g., *Pestalotiopsis*, *Phyllosticta* and *Septoria* spp.) are responsible for symptoms similar to those previously mentioned for foliar diseases, but are reported in carob orchards at a negligible frequency. Likewise, canker and branch diebacks caused by fungal species belonging to Botryosphaeriaceae are almost never recorded. Among the rots of wood tissues that may compromise old carob specimens, “brown cubical rot” caused by *Laetiporus sulphureus* is the most widespread and recurrent issue; this pathogen is also well-known for producing edible fruit bodies that are appreciated for pharmaceutical and industrial purposes. On the other hand, “white rots” caused by *Fomes* and *Ganoderma* species are less common and reported for the first time in this review. Gall-like protuberances on twigs of uncertain aetiology or tumors on branches associated with *Rhizobium radiobacter* are described, although these symptoms are seldom detected, as they are also observed for necrotic leaf spots caused by *Pseudomonas syringae* pv. *ciccaronei*. A worldwide list of pathogens not yet recorded but at high risk of potential introduction in Italian carob-producing areas is also provided. Finally, concerns related to new phytopathogenic fungi vectored by the invasive *Xylosandrus compactus* ambrosia beetle are addressed. All the described pathogens could become limiting factors for carob production in the near future, because they could be favored by high-density orchards, the increasing global network of trade exchanges, and the high frequency at which extreme events related to climate change occur globally. Thus, symptoms and signs, causal agents, epidemiology, and, whenever applicable, recommendations for disease prevention and management are provided in this review.

## 1. Introduction

Carob (*Ceratonia siliqua* L.) is an evergreen xerophyte tree belonging to the Fabaceae family and is mainly distributed in the Mediterranean basin [1]. Probably native to the Arabian Peninsula, the Horn of Africa, or the Middle East, carob has a long history spanning over 4000 years from the ancient Egyptians to the Cretans. Although the Greeks first recognized the agronomic and food value of carob pods, it was in the Middle Ages that Muslim Arabs first selected and spread the carob varieties that are currently being cultivated in Mediterranean countries [1,2,3]. Thereafter, carob has long been cultivated for human and animal consumption and, during famine periods, used as a valuable substitute for grain throughout South Europe, North Africa, the Middle East, and the Mediterranean Sea islands. Only in the 19th century was it introduced to the Southwestern USA (California and Arizona), Mexico, Hawaii, Chile, and Argentina by Spaniards; to the southern coast of Australia by Mediterranean emigrants; and to South Africa and India by the English [1,4]. Over the last few decades, the global agricultural economy has shifted towards more profitable crops, involving the partial replacement of the neglected carob crop [1,5,6]. 

Carob has more recently gained popularity due to the rediscovery of its value as a food, pharmaceutical, cosmetic, and biofuel resource and the related ecological advantages of the crop [7,8]. More specifically, the reawakened interest in crops that are best adapted to their domestication area, such as carob, is crucial for preserving biodiversity and stimulating sustainable agricultural practices [9]. The global carob market is projected to grow even further, with a compound annual growth rate (CAGR) of 4.55% and a predicted value of about USD 746 million by 2032 [4,10].

The carob crop has since spread and is being grown as a crop and utilized all around the world (Figure 1), including in Europe, Australia, Africa, the USA, and South America [4,8]. According to the Food and Agricultural Organization of the United Nations (FAO), the harvested area of the carob crop around the world is about 39,000 ha, corresponding to a global production of approximately 139,000 tons [11]. Almost all of the pod production is in countries within the Mediterranean region, i.e., Spain, Portugal, Italy, Morocco, Turkey, Greece, Cyprus, Algeria, and Lebanon, which are the foremost carob-producing countries (Figure 1). Italy’s harvested area is 5514 ha, with an annual production of 35,584 tons of carob pods [12]. However, evaluating the consistency of carob production potential in Italy and elsewhere in the Mediterranean region is difficult since plantings occur often within mixed orchards or as scattered trees in different agroforestry systems from which there was no harvest due to the inaccessibility of the crop site, damaged specimens, economic unfeasibility, and the abandonment of the plantations [4,13]. 

Typical features of carob such as drought, salt and atmospheric pollution resistance, and good adaptation to a wide range of dry soils have favored its widespread cultivation in the Mediterranean basin. Moreover, good tolerance to pests and pathogens may have contributed to ensuring the presence of numerous productive specimens. The lack of literature about carob diseases is noteworthy; the only information is in general textbooks about crops or in some local technical papers [1,14,15]. Likely, highly specialized agroecosystems for carob production, extreme events related to global climatic changes, and trade changes may promote the recrudescence of some diseases in epidemic form or the resurgence/occurrence of negligible or unknown diseases. 

Considering these factors, we report on the main and potential diseases of carob in the most representative Italian production area in the Mediterranean basin (Table 1). Symptoms and signs, causal agents, life cycle and, whenever possible, some recommendations for the prevention and management of the diseases are included.

## 2. Fungal Leaf Diseases

### 2.1. Powdery Mildew

*Pseudoidium* (syn. *Oidium*) *ceratoniae* (Comes) U. Braun and R.T.A. Cook is a fungus belonging to the family Erysiphaceae (Ascomycota) responsible for a carob disease known as carob powdery mildew. This disease is, together with Cercospora leaf spot, one of the two most important and cyclic diseases of carob in the Mediterranean area and worldwide. Powdery mildew develops characteristic symptoms and signs on leaves, flowers, and pods on both seedlings and mature plants, mainly in the late spring/early summer and autumn in Mediterranean countries (Figure 2A–J). 

Primary damage is recorded on leaves, where infection manifests as chlorotic spots with a typically powdery texture on the adaxial side of the leaf, corresponding to a greyish-white coloration on the abaxial side due to the production of mycelium and conidia. Over time, this turns to dark grey, almost black (Figure 2D–G). However, the symptom severity, and the related losses under stable ecological conditions, are usually negligible and economically acceptable. Under medium–high temperatures and scarce rainfall, powdery mildew infections can sporadically occur as an epidemic, causing heavy defoliation and significant reductions in both the yield and quality of carob production. However, adverse weather conditions can enhance leaf fall. The fungus, present in nature in conidial form, causes the formation of slightly discolored areas on both leaf blades. These areas have an initially smooth and sometimes rayed appearance. As the infection progresses, they become powdery and floury in consistency due to the presence of vegetative propagules (hyphae and conidiophores of the fungus) that form a white or greyish coating on the leaf surface. These spots can merge and, losing their white and powdery appearance, become dark and adopt a corky and reticulated appearance ((Figure 2A–G). When the fungal attack affects young leaves, the curling and waving of the leaf blades and edges may be observed. The leaves of *C. siliqua* are usually more susceptible during the growth stage, and infections are much less frequent when the leaves have become leathery. 

Similar spots to those described on the leaves can be found on infected fruits, although they are much less common. However, fruit infections may be slightly larger and more regular in shape. In the case of early and less severe attacks, the fruit growth stops, and the pods remain partially stunted. In the case of heavy attacks on young fruits, it is possible to observe a reduction in the pod number and size and poor fruit quality due to the lower sugar content in the pulp. In case of late attacks, the pathogen can occasionally cause serious damage to mature pods (Figure 2H,I) [14,16,17,18].

The fungus (first described as *Oidium ceratoniae* and reclassified as *Pseudoidium ceratoniae* in 2012) is a specialized obligate biotroph of the carob tree that can survive as mycelium in the tender tissues of the tree, such as leaves, fruit, buds, and twigs [18]. Similar to other powdery mildews, conidial production, propagule dispersal, and disease attacks are favored under high environmental humidity, although direct rainfall or excessive moisture is unfavorable for the infection process. The fungal infections occur preferentially within the 20–25 °C temperature range, although the optimum value may vary significantly depending on the fungal strain virulence and the carob cultivar. The shaded areas of the canopy can also promote powdery mildew attacks. Although no data about epidemiology are available in the literature about *Pseudoidium ceratoniae*, the disease cycle on carob is schematically summarized in Figure 3.

More severe leaf damage is recorded in specific cultivars. For instance, Negra, Melera, Costella, Santa Fe, and AA-2 are considered susceptible to powdery mildew infections, whereas Rojal, Matalafera, Amele di Bari, and Racemosa are quite resistant to disease. Sometimes, disease tolerance is due to a cultivar’s ability to compensate for the leaf drop in summer through adequate foliage, as is the case for Amele di Bari and Racemosa. Otherwise, this phenomenon is not observed for Saccarata and Giubiliana, which are therefore considered susceptible. Based on these considerations, a system for grading cultivar susceptibility to powdery mildew has been developed. Comprehensively, hermaphrodite cultivars (e.g., Alfieri and Tantillo) are even more tolerant to infections compared to those with female flowers. Typically, plants with male flowers are more susceptible to leaf infections occurring in fall, since they have poor vegetative recovery. Regarding fruit infections, it has been found that higher tolerance is related to the precocity and duration of flowering. For example, Racemosa and Giubiliana, which are late flowering cultivars, are considered tolerant [14,17,33,34]. 

The intervention thresholds are sufficiently high such that a lack of chemical applications is fully justified. Under a high level of disease pressure, more tolerant cultivars should be chosen for new carob plantings. Other control measures include: (i) adequate pruning to promote air circulation, especially in the shadiest portions of canopy trees; (ii) removing or burying infected leaves that have fallen to the ground; and (iii) the application of anti-powdery mildew fungicides (e.g., sulphur-based compounds) in spring, when primary infections occur.

### 2.2. Cercospora Leaf Spot

Similar to powdery mildew, Cercospora leaf spot is another fungal disease occurring worldwide in carob-producing countries, but the responsible fungus is considered to be native to the Mediterranean basin. The causal agent is the fungus *Pseudocercospora ceratoniae* (Pat. and Trab.) Deighton belonging to the family Mycosphaerellaceae (Ascomycota) [35,36], initially described as *Cercospora ceratoniae* in Algeria in 1903 [37]. The initial symptoms of the infection are characterized by small blackish-brown spots (2–3 mm) confined in the vein areas and, in advanced stages, surrounded by a chlorotic halo. Under humid conditions, the spots appear scattered throughout the leaf blade, becoming more numerous along the vein areas, and may merge into larger areas in both young and mature leaves (Figure 4A–D). *Pseudocercospora ceratoniae* forms greyish tufts on the leaflets; these structures are made up of dense bundles of conidiophores that emerge from the leaf stomata. Under high disease pressure made favorable by frequent rainfalls, the carob canopy appears severely damaged, and heavy defoliation is observed in orchards (Figure 4D) [14,19].

Since the sexual stage has not been yet detected, all publications to date have only reported infections caused by conidia formed in sporodochia, and an important role for ascospores in the disease cycle may only be hypothesized (Figure 5). The sporodochia develop in the lesions of leaves and fruits. The conidia are dispersed by splashes of raindrops and wind and are responsible for infections in the green and young tissues of the carob tree, which is the only known host of the pathogen [19]. Based on the above information, the Cercospora disease cycle may be represented as in Figure 5.

The recommendations for the management of Cercospora leaf spot are similar to those provided for carob powdery mildew.

### 2.3. Other Fungal Leaf Diseases

The carob canopy can occasionally be attacked by various fungal genera, which determine the foliar symptoms, characterized by dark coloration and thus called “black leaf spot”. Since these symptoms are similar to those of Cercospora, diagnosis in the field is very difficult, and the identification of the causal agent(s) should be performed in a laboratory by a certified plant pathologist. Although infections caused by these fungi are usually less severe than the main diseases above, an occasional epidemic, under particularly favorable conditions, can sometimes cause heavy damage (e.g., defoliation) in carob orchards. 

Among these phytopathogenic fungi, the most representative and frequent genera and species are *Alternaria alternata* (Fr.) Keissler, also responsible for Ceratonia blight [38], and some *Pestalotiopsis* species, i.e., *Pestalotiopsis uvicola* (Speg.) Bissett, *P. maculans* (Corda) Nag Raj and *P. biciliata* Maharachch., K.D. Hyde and Crous, which cause necrotic leaf spot surrounded by a dark halo [24,39,40]. Other fungi much less frequently associated with black leaf spot in carob are *Dothiora ceratoniae* (Quaedvl., Verkley and Crous) Crous, *Colletotrichum* spp., *Septoria ceratoniae* Pass., *Pileolaria ceratoniae* Rabenh., and *Phyllosticta ceratoniae* Berk. [14,15,23,41,42].

Comprehensively, these diseases, occurring alone or in association, currently have little or negligible practical importance for Italy.

### 2.4. Wood Decay Fungi

Many Basidiomycota (mostly Agaricomycetes) fungi are responsible for wood decay in many tree species, including carob, and for these reasons are known as “wood rot or decay fungi”. Wood rot is a process by which wood tissue disintegrates and can ultimately become friable and powdery. These fungi are able to utilize the cell wall and woody components thanks to their lytic enzymes. Some wood decay fungi are able to initiate the infection process, whereas others are considered weak pathogens; however, the consequent attacks can compromise the plant’s stability over a long period, affecting the physiological processes and wood characteristics and thus accelerating ageing processes. Carob, like other species such as olive or citrus, shows a certain degree of resilience against fungi causing heart rot. In Mediterranean conditions, it is possible to observe century-old specimens with a hollow trunk but still displaying a healthy canopy. One of the key factors for infection is tree age. The oldest specimens are usually the most susceptible to the attacks of these fungi. This is due to the high amount of dead woody tissue and the high likelihood of wound occurrence, which results in lesser vigor and, consequently, a weak host response to fungal attacks [14,21,43,44].

Some Agaricomycetes belonging to different genera discussed below are reported as causal agents of a few specific wood rots on carob in the Mediterranean basin.

#### 2.4.1. Brown Cubical Rot

*Laetiporus* Murrill (Polyporales) is a genus that includes many species growing on a wide host range of broadleaf and conifer trees, most of which are wood-decaying fungi having saprophytic and phytopathogenic behavior [20,45].

More specifically, *L. sulphureus* (Bull) Murrill is one of the most widespread species on fruit and forest trees [46,47,48], including carob, where it is responsible for the “brown cubical rot” [20,21] which is characterized by the production of shelf-shaped pale-yellow to pink-orange fruiting bodies that are rich in antioxidant and anti-microbial compounds appreciated for their nutritional and pharmaceutical value [49,50]. 

Fungal penetration occurs by means of basidiospores germinating on injured stems or roots by wounds, physiological cracks, or insect damage [46], and the fungus is responsible for the degradation of cellulose and hemicellulose but not lignin, which is only demethylated, leaving a brownish-appearing wood residue [44]. Fungus enters through natural and pruning wounds, physiological cracks, and insect damage and is defined as polyetic, since it completes its life cycle over more years, reaching the heartwood very slowly; it thrives on the infected or dead woody tissue, affecting larger portions of the tree. *Laetiporus sulphureus* develops decay symptoms longitudinally on the stem or branch rather than in the circumference direction (Figure 6A–F).

Described above are the signs of typical fruiting bodies, usually occurring between May and November under Mediterranean conditions. These fleshy fruiting bodies are located at the base of the trunk and have a semicircular shelf-like appearance, measure up to 40–50 cm, are overlapping, and have wavy edges [14,21]. 

As for its management, in the past it has been recommended to remove decayed tissues and fill the cavities with various materials. Nowadays, it is preferable not to remove portions of decayed tissues of carob tree due to the Compartmentalization of Decay in Trees (C.O.D.I.T.), in which pathogen progression is counteracted by means of forming more resilient tissues [21].

#### 2.4.2. White Rot and Other Wood Rot Fungi

Other fungi belonging to Agaricomycotina are able to act as pathogens of weakened and wounded trees by affecting cortical and wood tissues. For example, “white rot” caused by *Fomes* (syn. *Phellinus*) spp. and, more specifically, *F. igniarius* (L.) Fr., belonging to the family Polyporaceae, is occasionally reported in Italy and in the Mediterranean basin. In this type of rot, the wood gradually exhibits a fibrous and whitish appearance due to the degradation of all cell wall components. Symptoms appear a long time after penetration, and the carob tree undergoes progressive weakening due to the slow process of wood decay, which involves the partial emptying of the trunk and branches, which eventually die (Figure 7D). However, symptoms may be nonspecific, including the yellowing and poor vigor of the tree. The fruiting bodies, known as carpophores, show continuous growth and can persist for 15–20 years. They have a woody texture and a horseshoe-shaped form. The upper surface of these mature fruiting bodies is grey-black, initially smooth and later rough, while the marginal edges are brownish-yellow (Figure 7A–C). Infection typically occurs in the upper part of the plant through basidiospores and mycelium, and penetration takes place through pruning cuts and wounds, allowing the pathogen to spread within the woody cylinder. The management strategies are the same as those applied for *L. sulphureus* [14]. 

Other Agaricomycetes associated with wood alterations of the carob belong to the *Ganoderma* genus (Figure 7D,E), such as *G. lucidum* (Curtis) P. Karst. (Figure 7H), which was reported for the first time in Italy in this review, or to other fungi such as *Schizophyllum commune* Fr., which grows on dead stumps and branches [22]. As previously indicated, the adequate protection of pruning wounds on branches, avoiding stressful situations, the removal of infected tissue and pruning debris, and the use of healthy plant material are highly recommended for these Agaricomycetes.

### 2.5. Canker and Branch Dieback

These syndromes have been occasionally described in some Mediterranean countries where carob is cultivated. The typical symptoms are the progressive drying of shoots, branches, and twigs, and the infection always moves downward from the infection point. The wood of the affected branches shows a sectorial discoloration and necrosis with the occurrence of cankers. As the disease progresses, the branches dry up completely, involving the general decline and death of the tree. Regarding the carob tree, the only fungal species associated with this syndrome belong mainly to the Botryosphaeriaceae family [15,51]. More specifically, *Botryodiplodia aterrima* (Fuckel) Sacc., *Botryosphaeria* spp., and *Diplodia olivarum* A.J.L. Phillips, Frisullo and Lazzizera have been recorded in Italy [29,30,51].

### 2.6. Decay of Wood Roots

Some fungal pathogens, such as *Armillaria mellea* (Vahl. ex. Fr.) Kummer (Basidiomycota) and *Rosellinia necatrix* (Hart.) Berl. Prill. (Ascomycota), are potentially able to colonize the cambium of the thick roots of many woody crops, and they can also infect carob trees. These pathogens parasitize the root system over a long period, causing rot in the woody tissue and leading to the progressive weakening of fruit trees, which show reduced growth and development of the canopy and eventually die [52,53]. The management of these soil-inhabiting pathogens is very difficult due to the lack of authorized active molecules and, therefore, it should mainly rely on preventive approaches, taking into account the history of the previous crop, especially if it is a susceptible host. Whenever possible, the roots in the soil (rootlets) should be eliminated by successive cross-tillage passes to expose them on the surface, since dry and high-temperature conditions do not allow for the survival of these fungi in the debris. Few data are available and relate to past trials with the artificial inoculation of *Armillaria mellea* and *A. obscura*. In these conditions, young carob trees growing in a greenhouse have shown resistance to the root rot caused by *Armillaria* spp. infection [54]. The application of *Trichoderma*-based compounds can be generally effective in controlling *A. mellea* in fruit trees.

### 2.7. Verticillium Wilt

Although there is only one report on vascular wilt caused by *Verticillium dahliae* Kleb. in California, this serious disease affecting numerous herbaceous and woody hosts [51,55,56,57] could become a potential threat for carob; fortunately, it is not yet a concern for growers in Mediterranean countries. The only concern arises from the fact that olive trees are highly susceptible to Verticillium wilt, and the disease is quite widespread on olive plantings with which the carob trees are in contact in mixed agroecosystems [58,59]. Therefore, it is possible that the species attacking the olive trees may not be virulent towards carob.

### 2.8. Damping-Off

Damping-off syndrome is a common disease in nursery seedlings in a wide variety of hosts, including carob. The typical symptoms of these diseases are rot, peeling, and necrosis of the roots, as well as rot and necrosis in the neck of affected plants. As a consequence, there are aerial symptoms that include poor shoot development, chlorosis, wilting, defoliation, and the premature death of the plant. The causal agents associated with root rot and damping-off are oomycetes (*Phytophthora* spp., *Pythium* spp.) [51] and other fungal genera (e.g., *Fusarium*, *Rhizoctonia*, *Neonectria* spp.), most of which are well-known soil inhabitants, with high survival ability and a wide host range [60]. Nevertheless, these diseases have not yet been reported on carob seedlings, on the basis of confirmed aetiology, in surveys conducted in some carob nurseries in Italy.

## 3. Bacterial Diseases

### 3.1. Wood Galls of Uncertain Aetiology or Bacterial Cankers Caused by Rhizobium radiobacter?

The spread of galls on woody tissues is rarely observed in carob orchards in Italy. Specifically, these are irregularly shaped protuberances present on young branches and twigs, only located at the level of the bud cones. The morphology and regular distribution of these galls on the affected woody tissues do not support the hypothesis that their aetiology is due to bacterial infections such as those by the *Rhizobium radiobacter* (Beijerinck and van Delden, 1902) [26] [syn. *Agrobacterium tumefaciens* (Smith and Townsend, 1907) Conn 1942], as could be hypothesized based on the observation of macroscopic symptoms (Figure 8A,B). In this case, the alteration is most likely associated with physiological causes rather than biotic aetiology [61].

Nevertheless, galls and tumors affecting carob branches and stems that are caused by *R. radiobacter* have been officially recorded in other carob-producing countries [25,26,62]). This infection type is distinguished from the previous one by the formation of greater tumors consisting in a deep alteration of the tissue structure, which can be observed externally and consists of the occurrence of a tumor-like growth induced by the bacteria’s ability to transform cells into tumoral cells able to produce large amounts of indole-acetic acid. More recently these *R. radiobacter* infections were also observed in Sicilian carob orchards (Figure 8C,D).

### 3.2. Bacterial Leaf Infections

Bacterial leaf necrosis, caused by *Pseudomonas syringae* pv. *ciccaronei*, is a carob disease first described in Italy about 50 years ago. These bacterial infections are characterized by small necrotic leaf spots, often surrounded and/or accompanied by chlorotic halos, visible above all on the upper leaf blade (Figure 9A–C). More specifically, the causal agent *P. syringae* pv. *ciccaronei* was isolated and identified for the first time in the early 1970s from some Apulian carob orchards. The presence of this pathogen in the Mediterranean region is very sporadic, and its infections are favored by heavy rainfall and high humidity levels. In Sicilian carob orchards, the disease occurs in spring and stops during the summer [27,28]. Generally, the causative bacteria spread through their own exudates under high humidity conditions or free water, penetrating susceptible host tissue above all via the stomatal opening and any type of injuries. The wounds produced both in pruning and in harvesting are the means of entry for the pathogen. The control of this disease must be preventive, limiting the number of wounds and injuries, and as they occur, copper compound applications are recommended. Sanitation pruning and the elimination and destruction of pruning residues are necessary to prevent the survival and dispersion of the bacteria in the carob orchard.

## 4. Current Concerns: Potential Fungal Diseases Vectored by *Xylosandrus compactus*

The future of the carob crop in Italy and the Mediterranean basin is currently threatened by invasive *Xylosandrus compactus* Eichhoff, also known as the black twig borer, one of the most dangerous “ambrosia beetles”. The economic impact of this pest has increased in recent years, mainly due to global exchange and the higher frequency of extreme climatic events [63]. These factors generally reduce carob tree resilience to the black borer, which favors stressed trees [64]. The adult females burrow into the carob tree, excavating brood galleries in the xylem tissue and introducing the “ambrosia fungi”, which supply food for the larvae in galleries (Figure 10A,B). Among these, the true mutualistic fungi are carried into an insect structure called the “mycangium”, as is the case for *Ambrosiella xylebori* Brader (ex Arx and Hennebert), which provides food through fungus farming and is considered a “true” mutualist fungus [65,66]. Conversely, other fungi are carried on the external body surface of *X. compactus,* such as in the case of *Fusarium* spp., often belonging to the *Fusarium solani* species complex (FSSC), which shows pathogenic behavior, since it is involved in the development of necrosis symptoms on twigs and branches. Besides providing food for borer larvae, these fungi colonize the vascular tissue, cause the development of stem discoloration, block water and nutrient uptake, and eventually cause twig dieback [65,67,68,69]. Thus, the vectored fungus in this case contributes actively to exacerbating the pest damage. This phenomenon has been observed with increasing frequency in recent years in Italian carob orchards (Figure 10C,D).

One study on the biodiversity of the fungal community associated with *X. compactus* generation emphasizes the potential risk of introducing many exotic phytopathogenic fungi to new environments, including the carob agroecosystem [70]. This screening carried out on adults captured from *C. siliqua* showed that, among the recovered 60 fungal species, many of them were plant pathogens (Helotiales, Pleosporales, Hypocreales, and Diaporthales). In other words, these potentially phytopathogenic fungi could also occasionally be vectored on the external body of *X. compactus* and other ambrosia beetles, thereby contributing to the establishment of new threatening insect–fungus associations in many agroecosystems including carob [68,70,71]. For example, an increase in Fusarium dieback symptoms has been observed simultaneously to the heavy attacks of an invasive ambrosia beetle in many host plants in California [68]. Similarly, another study [72] showed that the attack by two ambrosia beetle species and an associated pathogenic oomycete, *Phytophthora ramorum*, infected coastal live oak trees, increasing the mortality of the trees by more than 50%. 

More recently, research has identified two new *Fusarium* sp. isolates belonging to the *Fusarium solani* species complex and vectored by *X. morigerus*. These strains displayed different pathogenicity abilities in different agroecosystems [73]. All these examples represent a definitive confirmation that ambrosia beetles could be vectors for other phytopathological issues.

## 5. Conclusions

This review reports on actual scenarios of recurrent phytopathological issues and future risk-introducing pathogens, with a focus on the challenges posed by exotic fungal pathogens that may also be vectored by *X. compactus* in Italy. This is a concern for Mediterranean carob growers since it has been demonstrated that the establishment of another exogenous biotic factor (e.g., pest) could make carob more susceptible to other diseases not previously considered key pathogens. Likewise, a possible change in an abiotic factor could result in a higher predisposition of carob trees to the above well-known or minor diseases. Consequently, the immediate quantification of climate change effects on the occurrence or recrudescence of diseases by means of forecasting models could be crucial for early epidemic detection and/or to establish whether carob plantings will be less or more profitable. Similarly, the continuous monitoring of the fungal community associated with *X. compactus* generations along their introduction and invasion pathways is of great relevance to avoid new associations that may cause direct or indirect harmful effects in the new carob agroecosystems.

Given the increasing interest in the carob crop, a deep understanding of pathogens and favorable factors should be encouraged to predict and simultaneously set up prevention and management approaches to phytopathological epidemics.

## Figures and Tables

**Figure 1 pathogens-12-01357-f001:**
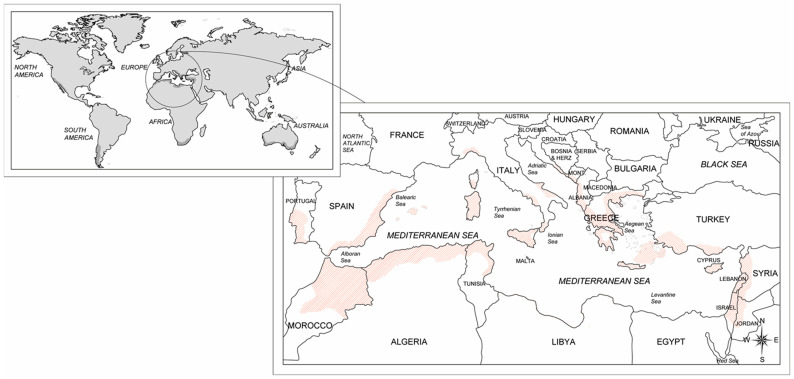
Global, Mediterranean, and Italian geographic distribution of the carob plantations.

**Figure 2 pathogens-12-01357-f002:**
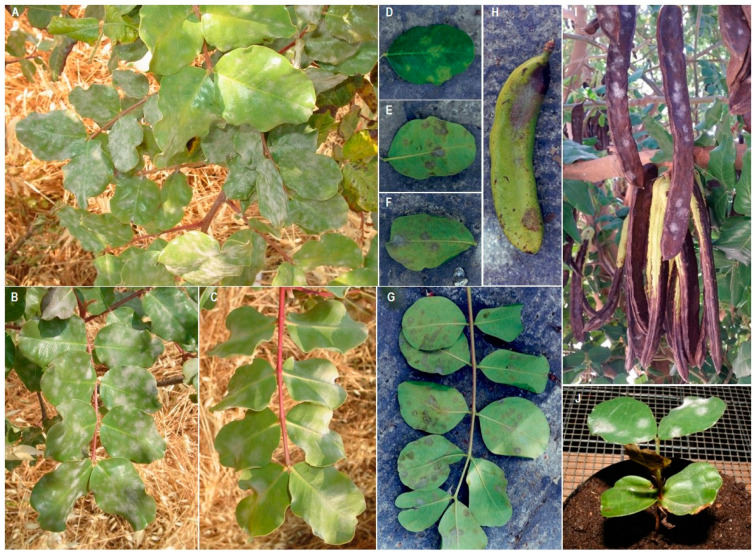
Infections caused by *Pseudoidium ceratoniae* on leaf blades of carob (**A**–**G**), young and mature pods (**H**,**I**), and carob seedlings (**J**).

**Figure 3 pathogens-12-01357-f003:**
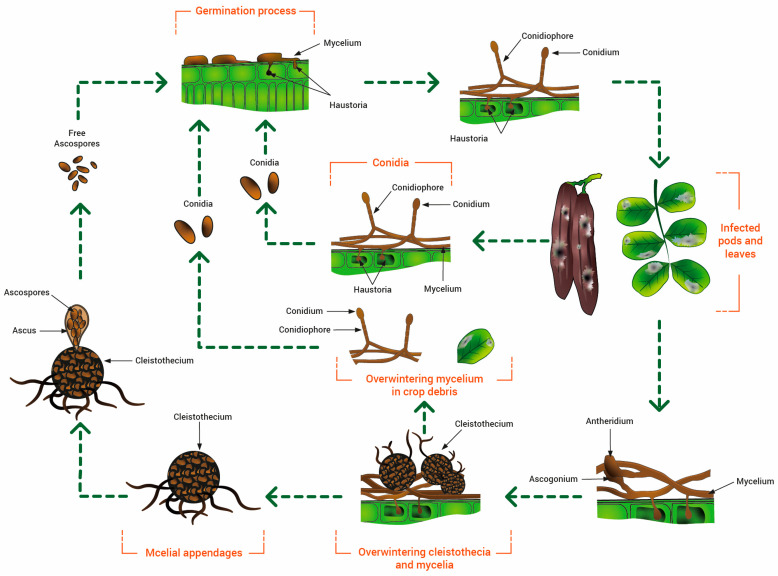
Disease cycle of *Pseudoidium ceratoniae* on carob (inspired by Agrios, 1997 [32]).

**Figure 4 pathogens-12-01357-f004:**
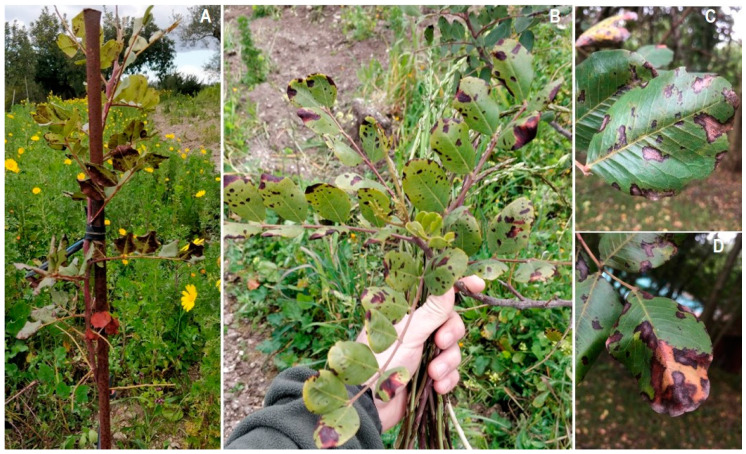
Infections caused by *Pseudocercospora ceratoniae* on carob. Symptoms on young carob seedlings (**A**,**B**); Cercospora infections on mature leaves causing heavy leaf drop (**C**,**D**).

**Figure 5 pathogens-12-01357-f005:**
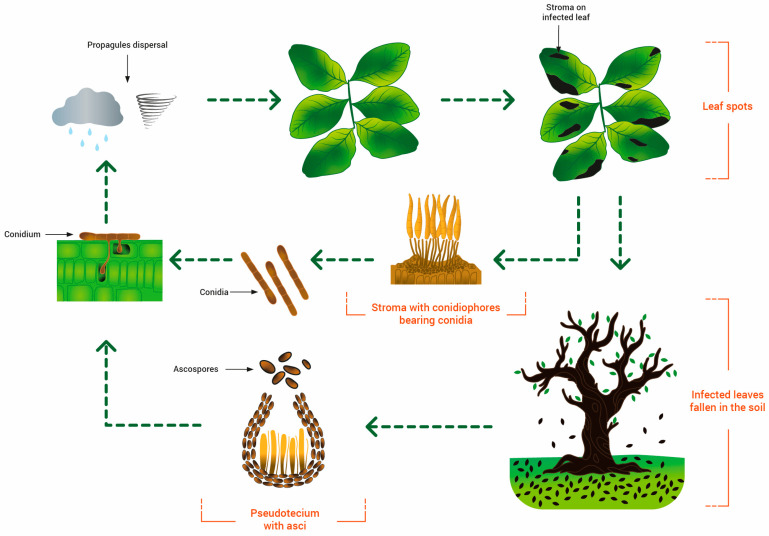
Life cycle of *Pseudocercospora ceratoniae* on carob.

**Figure 6 pathogens-12-01357-f006:**
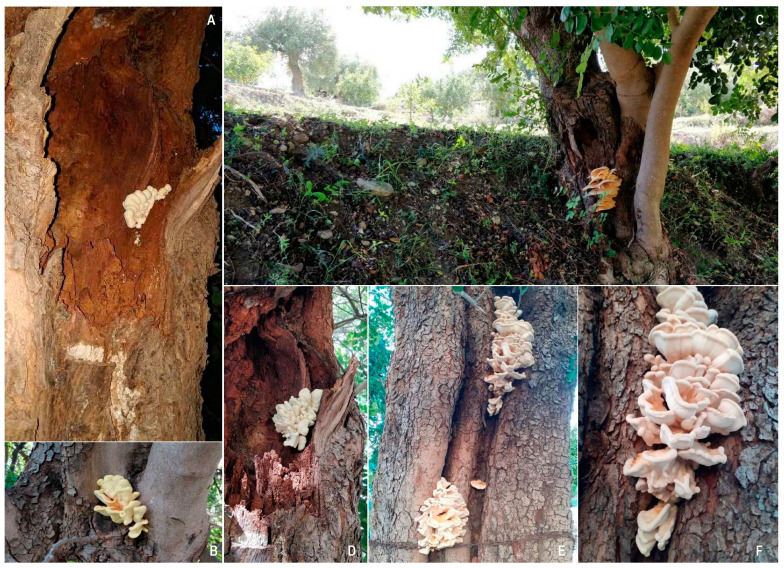
Infections of brown cubical rot on carob trunks caused by *Laetiporus sulphureus*. Fruiting bodies associated with longitudinal trunk rots (**A**–**F**).

**Figure 7 pathogens-12-01357-f007:**
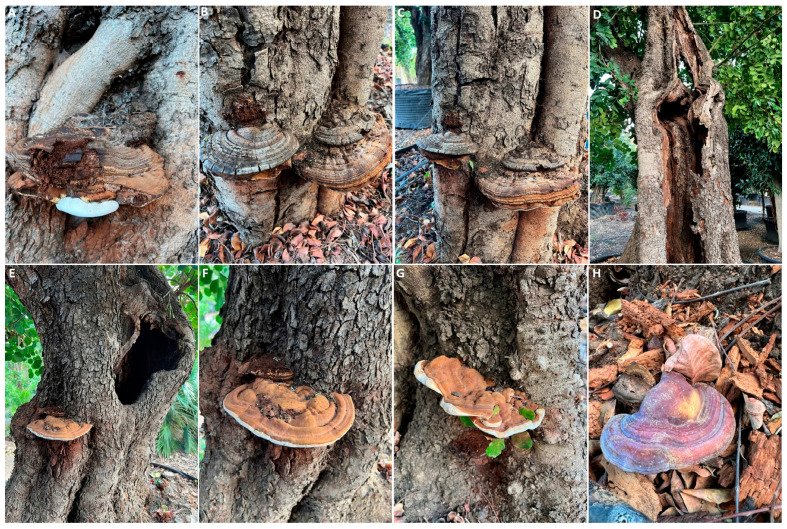
Wood decay infections and basidiomycetes fruiting bodies occurring on carob trunks and caused by *Fomes* spp. (**A**–**D**); *Ganoderma* spp. (**E**–**G**); and *Ganoderma lucidum* (**H**).

**Figure 8 pathogens-12-01357-f008:**
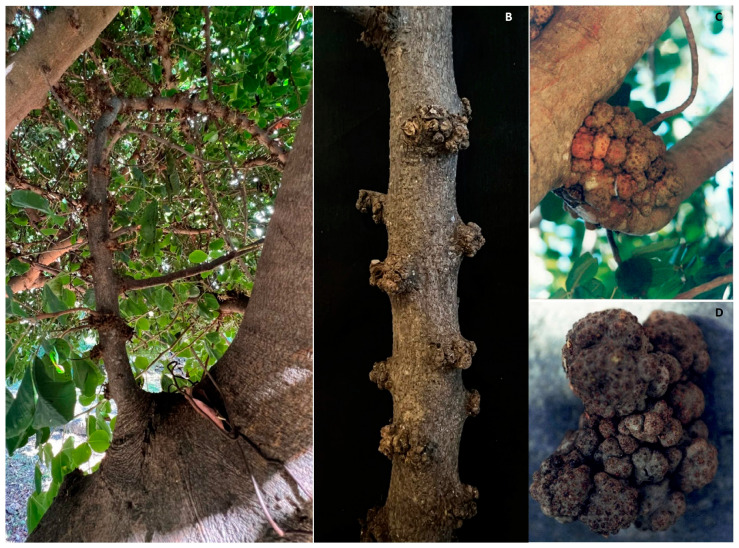
Occurrence of unknown aetiology galls at left and center (**A**,**B**) and tumors probably caused by *Rhizobium radiobacter* (**C**,**D**) on branches in Sicilian carob orchards.

**Figure 9 pathogens-12-01357-f009:**
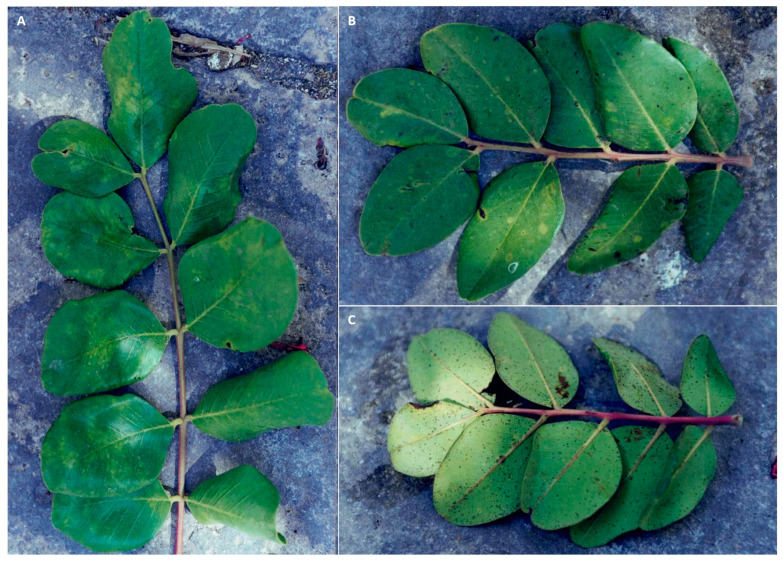
Symptoms of bacterial leaf spot caused by *Pseudomonas syringae* pv. *ciccarronei* (**A**–**C**) on carob leaves.

**Figure 10 pathogens-12-01357-f010:**
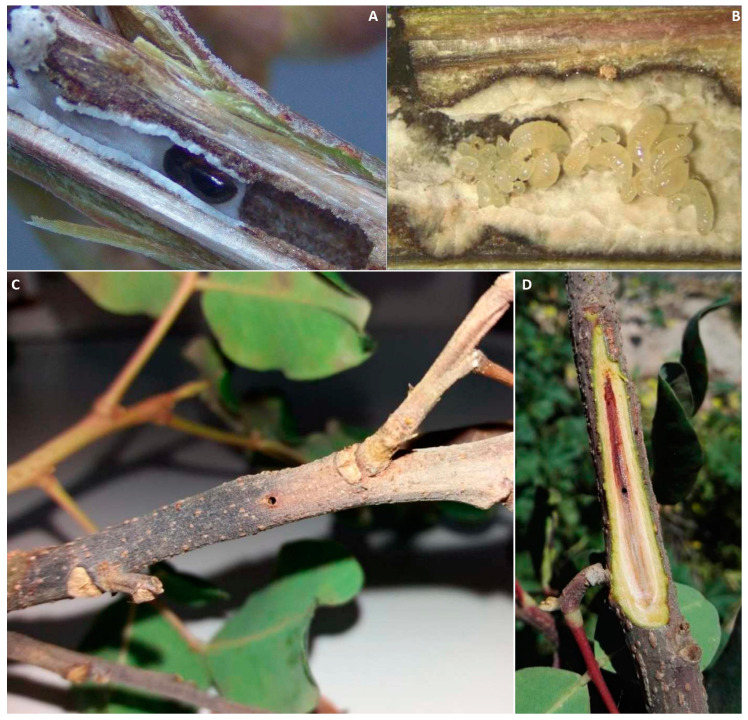
Gallery excavated by *Xylosandrus compactus* (**A**) and larvae and eggs on mutualistic fungal mycelium (**B**). Twig dieback and discoloration caused by *Fusarium solani* following black borer attacks (**C**,**D**) (Courtesy of Antonio Gugliuzzo).

**Table 1 pathogens-12-01357-t001:** List of phytopathological issues affecting carob trees grouped according to their current importance and diffusion in Italy.

Issue	Disease Common Name/Causal Agent(s)	Diffusion	References
Foliar diseases by fungi	Powdery mildew (*Pseudoidium ceratoniae*)	1	[14,16,17,18]
	Cercospora leaf spot (*Pseudocercospora ceratoniae*)	1	[14,19]
Wood decay or rot fungi	Brown cubical rot (*Laetiporus sulphureus*)	2	[14,20,21]
	White rot (*Fomes*, *Ganoderma,* and *Schizophyllum* spp.)	3	[14,22]; This review
Other fungal leaf diseases	Black leaf spot (*Pestalotiopsis* spp.; *Dothiora ceratoniae; Septoria ceratoniae*)	4	[14,23,24]
Bacterial diseases	Galls or tumors [*Rhizobium radiobacter*]	4	[25,26]; This review
	Bacterial necrotic leaf spot (*Pseudomonas syringae* pv. *ciccaronei*)	4	[14,27,28]
Canker and branch dieback	Shoot, branch, and twig dieback; cankers (*Botryodiplodia aterrima*, *Botryosphaeria dothidea,* and *Diplodia olivarum*)	4	[29,30]
Ambrosia fungi	*Fusarium solani* transmitted by *Xylosandrus compactus* *	1–5 *	[13,31]
Wood root rot	*Armillaria mellea* (Basidiomycota) and *Rosellinia necatrix*	5	-
Verticillium wilt	Vascular wilt (*Verticillium dahliae*)	5	-
Damping-off	*Phytophthora*, *Pythium*, *Fusarium*, *Rhizoctonia,* and *Neonectria* spp.	5	-

1 = diffuse; 2 = moderate diffusion; 3 = very low diffusion; 4 = very occasional; 5 = not yet reported. * Major current concern.

## Data Availability

Not applicable.

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
