# Peer review of "Major Pathogens Affecting Carob in the Mediterranean Basin: Current Knowledge and Outlook in Italy"

_pathogens, 2023, doi:10.3390/pathogens12111357_

Round 1
Reviewer 1 Report
Comments and Suggestions for Authors
please see comments in attached file

Author Response
Reviewer 1 - Comments and Suggestions for Authors
Review: Main carob pathogens in the Mediterranean basin: current knowledge and outlook in Italy.
Comments:
2.1 Powdery Mildew
Q1. Ln 97 – what is the primary damage type on carob?
A1. Thank you. To this regard we clarified with some modifications/adjustments in the MS. However, authors describe primary damages in the following sentences as “…with medium-high temperature….heavy defoliations implying a decrease of yield and quality of production” (Ln 103-105) and less frequent damages on infected fruit “…early……and severe attacks on young fruits and late attacks on mature pods” (Ln 116-121).
Q2. Ln 106 – which are the most important environmental conditions to disease severity?
A2. Ok. Although epidemiological data are not available in literature, the disease usually occur in Mediterranean orchards during late spring/early summer and autumn seasons at temperature of about 20-25°C, environmental humidity higher than 70% (although direct rainfall or excessive moisture are unfavourable for disease). Moreover, the disease is favoured by partial shading (see revised text)
Q3. Ln 133 – perhaps a table of the the things that contribute to disease?
A3. Thank you. Although authors do not provide a table we summarized in a better manner the factors conducive for the powdery mildew infections in carob orchards (see the revised MS).
Q4. REVIEWER wonders how testing and grading powdery mildew susceptibility determines best cultivar and do you have a suggested cultivar for Italy?
A4. Thank you. The authors provided herein only some information about cultivar susceptibility that could be useful for new orchard establishments. However, the cultivar in old orchards varied according to geographic area in Italy (e.g. above Amele di Bari in Apulian area and Latinissima in Sicilian area)
2.2. Cercospora leaf spot
Q5. Ln 168-170 … consider rewriting sentence… “Like powder mildew, Cercospora leaf spot is a fungal pathogen widely distributed in all carob producing countries, although it is considered native to Mediterranean basin. Ln – 195-197 … Based on above information and hypothesizing the occurrence of no known telemorph…
A5. Thank you. We performed the modifications (see revised MS).
2.3. Other fungal leaf diseases
Q6. Ln 208-209 … add the following to the end of 209… by a certified plant pathologist.
A6. ok done. (see rev MS).
Q7. REVIEWER would like clarification with 222 and 223, are you implying little potential for damage in Italy?
A7. Correct, for Italian geographic area. (Modification done)
2.4.1. Brown cubical heart rot
Q8. Ln 271- consider: … Previously it had been recommended to remove decayed tissues and fill cavities with various materials. (continuing): … Today, it is preferable not to remove portions of decay due to Compartmentalization of Decay in Trees (C.O.D.I.T.) in carob trees.
A8. Thank you. We accordingly modified the sentence.
2.6. Wood root rot
Q9. REVIEWER wonders if a more clarified impact to carob beyond potentially could be expressed. For example a rated system of severity; low, medium, high?
A9. Unfortunately, a detailed insight on this carob disease is nowadays not possible. Data reported in overview refer to preliminary data obtained from experiment performed under artificial inoculation conditions.
2.7. Verticillium wilt
Q10. REVIEWER wonders whether a management recommendation on the mixed agroecosystems could be brought forward to help move away from risk of Vert via olives?
A10. Ok. The management should refer to the accurate selection of propagation material for example of olive or other trees host susceptible to disease.
3.2. Bacterial Leaf Infection
Q11. REVIEWER wonders if you have any data (413) to demonstrate the increase of X. compactus in Italy?
A11. The authors do not have exact numerical data, but it can be certainly assumed that issue has interested many important production district (Ragusa and Syracuse provinces) in the Sicilian area, where serious damages are being registered in many commercial orchards
Q12. FINAL. Well written paper with few to minimal changes needed. Perhaps from this could come a practitioners guide to management beyond just knowledge presented and be the start of further review of the research gaps.
A12. Thank for your final comment and for your precious efforts to improve and made more readable this review. I agree with your final comment since the overview/scientific guide to carob diseases and management was the initial objective of the authors.
Reviewer 2 Report
Comments and Suggestions for Authors
This paper is a much-needed review of diseases currently known to cause problems in carob production, focused on Italy. Carob is, arguably, increasing in importance as a crop, with the bean flour being more popular for alternative lifestyles. It is also grown for food in most Mediterranean countries. I would like to see the paper published, but it needs serious language editing before it can be considered suitable for any journal.
The title is not 'good English'. You are describing the major pathogens affecting carob in the Mediterranean basin.
Lines 15-18: are heart rots really 'diseases'. It's a little ambiguous, as the decay fungi are attacking dead wood, even though that wood is (sometimes) within the tree itself. Also, Laetiporus sulphureus causes a brown cubical rot: no need to include 'heart'. You address these points to some extent in the main text of the paper.
Line 19: White rots, not 'white decays'. [also requires attention in Table 1]
Line 21: spelling of Rhizobium.
Line 93: using an old taxonomic approach with 'Ascomycetes'. Use a more up-to-date taxonomy.
Line 95: 'cercosporiosis' is not a good name for the next disease listed. It is very out of date.
Lines 105-107: implies that the disease is bad, even when not present?! Separate the effects of the pathogen from the impact of severe drought.
Line 143: Is the figure a generalised powdery mildew life cycle, modified from Agrios to represent carob mildew?
Line 185-186: Figure 4D shows nothing at all - it looks like a photograph of a random patch of soil.
Line 224: 'Wood decay fungi
Line 225: Again, 'basidiomycetes' is an old term for these fungi. They considered to be Agaricomycetes (mostly).
Lines 228-232: these statements are not strictly correct. Some wood decay fungi ARE able to initiate infections - Armillaria mellea is probably the best know example, but other species are also capable of that process. I suspect the authors are writing about fungi which commonly result in stem decays, which are assumed to infect through wounds (although even that assumption may not be correct).
Lines 233-235: Again, this is a problem of interpretation. The fungi the authors are writing about are heart rotting species, which attack and digest the dead heartwood. Sapwood remains alive and is unaffected, hence a tree is able to continue to grow well, despite the heart rot.
Line 244: Brown cubical rot.
Line 245: now the authors use the more up-to-date 'Basidiomycota'!
Line 268: Are pores of Laetiporus a 'few mm wide'?
Line 278 (and elsewhere): They are white rot fungi. Calling it 'white decay' is rather clumsy.
Line 292-294: check the infection biology of Fomes spp. Why is it via mycelium and not basidiospores?
Line 319: Decay of woody roots
Lines 339-345: it is possible that 'strains' of Verticillium dahliae that infect olive are unable to infect carob.
Line 354: I suspect that, to date, little work has been carried out on possible Phytophthora infections of carob. Please be careful in the text not to suggest that oomycetes are fungi.
Lines 362-380: Much of the text on bacterial cankers is very difficult to understand.
Lines 386 -402: I am surprised that this foliar bacterial pathogen is considered to enter via wounds. Wounds would probably be fairly uncommon on the foliage (in the absence of serious aphid or similar insect) attack. Does the pathogen enter through stomata under high humidity, perhaps?
Lines 448-449: there was no particular focus on pathogens vectored by Xylosandrus - it received the same emphasis as most of the other damaging examples.
Comments on the Quality of English Language
The main problem with this manuscript is the poor quality of the English language used. During revision, the authors MUST get the help of a native English speaker who has experience in plant pathology, otherwise much of the meaning will be unclear or obscure to readers.
Please do not use the phrase 'the so-called' before phrases - it is not necessary. Ever.
Author Response
Reviewer 2 - Comments and Suggestions for Authors
Q1. This paper is a much-needed review of diseases currently known to cause problems in carob production, focused on Italy. Carob is, arguably, increasing in importance as a crop, with the bean flour being more popular for alternative lifestyles. It is also grown for food in most Mediterranean countries. I would like to see the paper published, but it needs serious language editing before it can be considered suitable for any journal.
A1. Thank you for the critical comment and for your efforts to improve this overview. However, the authors provided language editing by a specialist (see attached certificate).
Q2. The title is not 'good English'. You are describing the major pathogens affecting carob in the Mediterranean basin.
A2. Thank you. The authors changed the title according to your suggestions
Q3. Lines 15-18: are heart rots really 'diseases'. It's a little ambiguous, as the decay fungi are attacking dead wood, even though that wood is (sometimes) within the tree itself. Also, Laetiporus sulphureus causes a brown cubical rot: no need to include 'heart'. You address these points to some extent in the main text of the paper.
A3. Thank you for the comment. The authors addressed here and through entire MS and performed accordingly many modifications (see revised MS).
Q4. Line 19: White rots, not 'white decays'. [also requires attention in Table 1]
A4. Ok done. modifications performed.
Q5. Line 21: spelling of Rhizobium.
A5. Ok, done. Thank to your indication the authors discover another typos/refuse/mistake (e.g. Pestalotiopsis)
Q6. Line 93: using an old taxonomic approach with 'Ascomycetes'. Use a more up-to-date taxonomy.
A6. Ok, done. (see revised MS)
Q7. Line 95: 'cercosporiosis' is not a good name for the next disease listed. It is very out of date.
A7. You are correct. We have replaced this term with Cercospora leaf spot.
Q8. Lines 105-107: implies that the disease is bad, even when not present?! Separate the effects of the pathogen from the impact of severe drought.
A8. I agree. It is a very confuse sentence. I modificate the sentence (see revised MS).
Q9. Line 143: Is the figure a generalised powdery mildew life cycle, modified from Agrios to represent carob mildew?
A9. Exactly a Figure drawn from Agrios inspired the authors for the realization of the original colour figure (last version). This cycle includes “ex novo” images and items. The conceptualization and images rendering were inspired by how powdery mildew symptoms appear and as the disease progresses.
Q10. Line 185-186: Figure 4D shows nothing at all - it looks like a photograph of a random patch of soil.
A10. You are correct. It is unclear. This picture refers to direct consequence of heavy disease attack with many carob leaves fallen in the soil. However, the authors replace this photo.
Q11. Line 224: 'Wood decay fungi
A11. Ok, done
Q12 Line 225: Again, 'basidiomycetes' is an old term for these fungi. They considered to be Agaricomycetes (mostly).
A12. Ok done
Q13. Lines 228-232: these statements are not strictly correct. Some wood decay fungi ARE able to initiate infections - Armillaria mellea is probably the best know example, but other species are also capable of that process. I suspect the authors are writing about fungi which commonly result in stem decays, which are assumed to infect through wounds (although even that assumption may not be correct).
A13. Thank you. We have modified the sentences according to your comments
Q14. Lines 233-235: Again, this is a problem of interpretation. The fungi the authors are writing about are heart rotting species, which attack and digest the dead heartwood. Sapwood remains alive and is unaffected, hence a tree is able to continue to grow well, despite the heart rot.
A14. Thank you. The authors rephrase these sentences according to your suggestions and comments
Q15. Line 244: Brown cubical rot.
A15. Ok. Done
Q16. Line 245: now the authors use the more up-to-date 'Basidiomycota'!
A16. Done. see revise MS
Q17. Line 268: Are pores of Laetiporus a 'few mm wide'?
A17. Yes. However, we deleted this sentence. It provides nothing information.
Q18 Line 278 (and elsewhere): They are white rot fungi. Calling it 'white decay' is rather clumsy.
A18 Ok. We have replaced the term through entire MS.
Q19 Line 292-294: check the infection biology of Fomes spp. Why is it via mycelium and not basidiospores?
A19 thank you once again. I checked and modified accordingly the sentence (see revised MS)
Q20. Line 319: Decay of woody roots
A20. Done
Q21. Lines 339-345: it is possible that 'strains' of Verticillium dahliae that infect olive are unable to infect carob.
A21. It is quite probably
Q22. Line 354: I suspect that, to date, little work has been carried out on possible Phytophthora infections of carob. Please be careful in the text not to suggest that oomycetes are fungi.
A22. You are correct. In the text we provided the modifications to give right information
Q23. Lines 362-380: Much of the text on bacterial cankers is very difficult to understand.
A23. The authors improved editing language of this section. See
Q24. Lines 386 -402: I am surprised that this foliar bacterial pathogen is considered to enter via wounds. Wounds would probably be fairly uncommon on the foliage (in the absence of serious aphid or similar insect) attack. Does the pathogen enter through stomata under high humidity, perhaps?
A24. you are correct. We have omitted the main penetration via of this pathogen. However, the authors provide the right modifications
Q25. Lines 448-449: there was no particular focus on pathogens vectored by Xylosandrus - it received the same emphasis as most of the other damaging examples.
A25. The authors implemented this section.with other examples.
Q26. The main problem with this manuscript is the poor quality of the English language used. During revision, the authors MUST get the help of a native English speaker who has experience in plant pathology, otherwise much of the meaning will be unclear or obscure to readers.
A26. We provided to an extensive a language editing by a speciailist.
Q27. Please do not use the phrase 'the so-called' before phrases - it is not necessary. Ever.
A27. Ok. we deleted the term everywhere.
Yours Sincerely
Alessandro Vitale

Round 2
Reviewer 2 Report
Comments and Suggestions for Authors
Despite the ‘language certificate’ there remain many small errors in the manuscript, which is unfortunate. Please get the manuscript checked again.
Line 100: ‘on the down blade…’ – better to refer to the axial and adaxial sides of the leaf.
Line 167: ‘ascomycete’ is used here, instead of up-to-date taxonomy.
Line 199: delete moreover here. It is a word used to link information in two sentences.
Line 287: I would only expect to see Schizophyllum commune on branches that have been dead for several years.
Line 290: ‘use of healthy plant material is highly recommended for these basidiomycetes.’ The sentence mentions several approaches and, hence, is plural. Please replace ‘basidiomycetes’ with Agaricomycetes.
Line 350: replace ‘portions’ with ‘tissues’.
Lines 431-432: Phytophthora ramorum is NOT a fungus!
Line 456: use ‘epidemics’.
Comments on the Quality of English Language
I have reviewed the manuscript again and it still requires language editing.
Author Response
Reviewer 2 - Comments and Suggestions for Authors
Q1. Despite the ‘language certificate’ there remain many small errors in the manuscript, which is unfortunate. Please get the manuscript checked again.
A1. I regret. However, I have upload the re-edited review for further language check.
Q2. Line 100: ‘on the down blade…’ – better to refer to the axial and adaxial sides of the leaf.
A2. Thank you for the comment. I replace the terms with adaxial and abaxial side of the leaf, respectively (see MS Rev.2)
Q3. Line 167: ‘ascomycete’ is used here, instead of up-to-date taxonomy.
A3. Modified. See revised manuscript.
Q4. Line 199: delete moreover here. It is a word used to link information in two sentences.
A4. Thank you. Done.
Q5. Line 287: I would only expect to see Schizophyllum commune on branches that have been dead for several years.
A5. You are correct. The sentence is incomplete. I modify it in the revised manuscript (see annotated pdf).
Q6. Line 290: ‘use of healthy plant material is highly recommended for these basidiomycetes.’ The sentence mentions several approaches and, hence, is plural. Please replace ‘basidiomycetes’ with Agaricomycetes.
A6. Ok. Done.
Q7. Line 350: replace ‘portions’ with ‘tissues’.
A7. Ok done
Q8. Lines 431-432: Phytophthora ramorum is NOT a fungus!
A8. I’m very sorry for this unforgivable OVERSIGHT due to excessive hurriedness. Once again, thank you very much.
Q9. Line 456: use ‘epidemics’.
A9. Ok. Done.
Yours Sincerely
Alessandro Vitale
